# A Typical Case Presentation with Spontaneous Visual Recovery in Patient Diagnosed with Leber Hereditary Optic Neuropathy Due to Rare Point Mutation in *MT-ND4* Gene (*m.11253T>C*) and Literature Review

**DOI:** 10.3390/medicina57030202

**Published:** 2021-02-26

**Authors:** Rasa Liutkeviciene, Agne Sidaraite, Lina Kuliaviene, Brigita Glebauskiene, Neringa Jurkute, Lina Aluzaite-Baranauskiene, Arvydas Gelzinis, Reda Zemaitiene

**Affiliations:** 1Department of Ophthalmology, Lithuanian University of Health Sciences, Medical Academy, Eiveniu 2 str., LT-50161 Kaunas, Lithuania; agnesidaraite@gmail.com (A.S.); lina.kuliaviene@gmail.com (L.K.); bglebauskiene@gmail.com (B.G.); aluzaite.lina@gmail.com (L.A.-B.); Arvydas.Gelzinis@kaunoklinikos.lt (A.G.); Reda.Zemaitiene@kaunoklinikos.lt (R.Z.); 2Neuroscience Institute, Lithuanian University of Health Sciences, Medical Academy, Eiveniu str. 2, LT-50161 Kaunas, Lithuania; 3Genetics Department, Moorfields Eye Hospital, 162 City Road, London EC1V 2PD, UK; jurkute.neringa@gmail.com

**Keywords:** Leber hereditary optic neuropathy, LHON, aetiology, diagnosis, treatment

## Abstract

Leber hereditary optic neuropathy (LHON) is one of the most common inherited mitochondrial optic neuropathies, caused by mitochondrial DNA (mtDNA) mutations. Three most common mutations, namely *m.11778G>A*, *m.14484T>G* and *m.3460G>A*, account for the majority of LHON cases. These mutations lead to mitochondrial respiratory chain complex I damage. Typically, LHON presents at the 15–35 years of age with male predominance. LHON is associated with severe, subacute, painless bilateral vision loss and account for one of the most common causes of legal blindness in young individuals. Spontaneous visual acuity recovery is rare and has been reported in patients harbouring *m.14484T>C* mutation. Up to date LHON treatment is limited. Idebenone has been approved by European Medicines Agency (EMA) to treat LHON. However better understanding of disease mechanisms and ongoing treatment trials are promising and brings hope for patients. In this article we report on a patient diagnosed with LHON harbouring rare *m.11253T>C* mutation in *MT-ND4* gene, who experienced spontaneous visual recovery. In addition, we summarise clinical presentation, diagnostic features, and treatment.

## 1. Introduction

Leber hereditary optic neuropathy (LHON) is one of the most frequent mitochondrial disorders and hereditary optic neuropathies. It is caused by mitochondrial deoxyribonucleic acid (DNA) mutation. The three most common point mutations result in damage of the mitochondrial respiratory chain complex I. LHON affects young individuals (usually from the age of 15 to 35) with male predominance [1]. Acute, painless bilateral loss of central vision is the main symptom of this disease which makes young people legally blind (*V* < 0.05) within a few months from disease onset. The loss of vision is usually irreversible, however cases describing spontaneous vision recovery have been reported [2]. In 2014, 120 new cases of LHON were diagnosed in Japan, the population of which is estimated to be approximately 127 million [3]. According to various meta-analysis from Europe, the prevalence of LHON is about 1:45,000, while the incidence of LHON is 1:1,000,000 [2].

## 2. Aetiology

LHON is caused by mutations occurring in the mitochondrial DNA. The three most common mtDNA point mutations are: *m.3460G>A MT-ND1* (5–10%), *m.11778G>A MT-ND4* (50–70%), and *m.14484T>C MT-ND6* (15–30%) [4,5]. The three most common variants account for 90% of the LHON cases with remaining 10% caused by other rare variants Based on the data of MitoMap (http://www.mitomap.org/MITOMAP (accessed on 25 February 2021)) there are 16 more mtDNA mutations related to LHON and 18 more mutations that are not yet proven to cause LHON (there is a lack of cases to prove pathogenicity).

Compared to classical Mendelian disorders, mitochondrial diseases genetics is complex with three main features namely, maternal inheritance, random segregation and heteroplasmy levels. Mitochondrial DNA (mtDNA) is inherited through the maternal line as human sperm cell has only a few mitochondria which does not contribute greatly to the zygote. As a result, a child’s mtDNA with mutations will be inherited exclusively only from the mother [1]. A single mitochondrial defect would lead to heteroplasmy. As mitochondria randomly segregates it might lead to different levels of heteroplasmy or even homoplasmy in cell. It is an important feature to understand as if it reaches a specific threshold, it can lead to disease or account for its severity.

Indeed, LHON is complex disease with multifactorial genetic and environmental triggers interacting with the LHON mtDNA mutation [6]. Many studies have provided convincing evidence that heavy smoking, and to a lesser extent excessive alcohol consumption, could lead to visual loss in LHON mutations carriers [7,8,9,10,11]. Obviously, LHON could arise because of hormonal differences, especially the higher levels of circulating estrogens in women. This would imply a second peak of female LHON carriers converting in the perimenopausal or menopausal period, but there is no robust data yet to suggest that this is the case. Retinal ganglion cells (RGCs) express high levels of the estrogen beta receptor and estrogen derivatives are known to exert a neuroprotective effect under conditions of heightened cellular stress [12,13].

## 3. Clinical Features

LHON is one of the mitochondrial diseases which in majority of cases exclusively affects eye structures. However, a proportion of cases has been reported to express additional neurological symptoms [14,15,16] or white matter lesions, similar to multiple sclerosis (MS) [17,18]. In such cases, the disease is called LHON plus syndrome.

### 3.1. Ocular Symptoms


Functional:
Bilateral, painless subacute failure of central vision. In majority of cases, the second eye is usually affected within the period of 12 months (the average is 3–4 months) since initial presentation. Bilateral concurrent visual loss occurs in about 25 percent of the patients. In most cases visual acuity ranges between counting fingers to perception of light which make patients legally blind (*V* < 0.05). Spontaneous partial visual regression is possible in rare cases and it is more likely to happen for the patients who have *m.14484T>C* mutation than for those with *m.11778G>A* or *m.3460G>A* pathogenic variant [2]. Other factors of better prognosis were estimated, such as earlier onset of LHON (age < 10 years), subacute manifestation of LHON with slow deterioration of visual acuity and a relatively large optic nerve disc [19,20,21].Loss of colour vision affecting mostly the red-green system [20].Visual field testing shows dense central or centrocecal scotoma [21,22].Reduced contrast sensitivity.Electrophysiological studies: pattern electroretinography (pERG) and visual evoked potentials (VEP) show dysfunction of the optic nerve and the retinal ganglion cells [23]. Some studies show that recording photopic negative response (phNR) could be an informative test in LHON [24].Structural changes:Each stage of LHON (described below) has characteristic features of fundus appearance.


### 3.2. Extraocular Symptoms and Signs (LHON Plus Syndrome)

Neurological signs:Postural tremorPeripheral neuropathiesMovement disordersMultiple sclerosis-like illness (Harding syndrome: MS and LHON symptoms occur together)Nonspecific myopathyCardiac arrhythmias

### 3.3. Other Neurological Signs Which Are Associated with LHON

Some neurological signs for example, postural tremor, peripheral neuropathies, nonspecific myopathies, movement disorders, Leigh syndrome were noticed to be common in LHON patients [25,26]. Some individuals with LHON, usually women, may develop a progressive multiple sclerosis (MS) like illness [27]. The loss of vision is different from the classic LHON: recurrent episodes of visual loss that can be associated with ocular pain occur. After each such episode loss of vision regresses only partially and this eventfully leads to total blindness for half of the patients [27]. Also, with bilateral optic neuropathy, disseminated central nervous system demyelination, characterized by periventricular white matter lesions, develops for these patients but oligoclonal bands in the cerebrospinal fluid are absent [28,29,30].

## 4. Stages of LHON

According to consensus, LHON has the following stages:Asymptomatic (the carriers of the mutation). Using optical coherence tomography (OCT) imaging, thickening of the temporal retinal nerve fibre layer was confirmed in asymptomatic individuals with a LHON—causing mtDNA pathogenic variant, providing evidence that the papillomacular bundle is particularly vulnerable in LHON [31].Subacute (from the manifestation of LHON to 6 months). Patients are usually asymptomatic until they develop visual blurring affecting the central visual field in the first eye (acute phase). Similar symptoms appear in the other eye an average of two to three months later, so that both eyes are affected in the majority of cases within six months. Unilateral damage of the optic nerve is very rare in patients with LHON and in such cases another underlying pathologic process should be actively excluded. The most common characteristic is an enlarging central or centrocecal scotoma and as the field defect increases in size and density, visual acuity deteriorates to the level of counting fingers or worse. In this stage, even for the asymptomatic patients, specific alterations of the fundus could be observed: retinal telangiectasia of the peripapillary vessels, edema of the peripapillary retinal nerve fibre layer, which progress, and eventually atrophy of optic nerve occurs [5].Dynamic (from months 6 to 12). Alterations of the fundus, which occur in the subacute stage, are slowly regressing: the edema of the peripapillary retinal nerve fibre layer is decreasing.Chronic (more than 12 months). In this stage, optic nerve atrophy is progressing at various rates: it could be observed from 6 weeks to more than 1 year after the loss of vision. Central or centrocecal scotoma is enlarging. Most patients remain severely visually impaired and are within the legal requirements for blind registration [32]. OCT shows the retinal fibre layer thinning, especially in temporal zones.

## 5. Diagnostics

LHON diagnosis is established when ophthalmological symptoms (or signs) are recognized and/or one of the mitochondrial DNA pathogenic variants are diagnosed by molecular genetic tests. It is important to exclude other possible cause of optic neuropathy (Figure 1).

Genetic testing:

Targeted gene panel test. The three most common mtDNA pathogenic variants consist 90–95% of those causing LHON. Targeted testing is the first test choice for searching one of these three variants.
*m.3460G>A*, in *MT-ND1*.*m.11778G>A* in *MT-ND4*, present in 70% northern European and in 90% Asian patients [36,37,38].*m.14484T>C* in *MT-ND6*, commonly found among French Canadians due to a founder effect [39,40,41].A multi-gene panel that includes the mitochondrial genes namely, *MT-ND1*, *MT-ND2*, *MT-ND4*, *MT-ND4L*, *MT-ND5*, and *MT-ND6,* that encode subunits of NADH dehydrogenase. The included genes and the sensitivity of multi-gene panels vary by laboratory and over time. Performing this test, not only the most frequent but also rare pathological variants could be determined [42].Whole mtDNA sequencing if pathogenic variant has not been identified during previous genetic testing.

## 6. Treatment

Multiple LHON treatment trials were carried out in order to find the best treatment option for LHON. This includes various neuroprotective agents, stem cell therapy, infrared radiation therapy and even gene therapy [43,44,45,46,47,48,49,50] (Table 1). The most promising of these methods are gene and stem cell therapies but further research is needed to evaluate their effectiveness [51,52].

**Table 1 medicina-57-00202-t001:** Other non-approved treatment modalities for LHON.

Therapeutic Options	Mechanism of Action	Findings in Clinical Studies
1. Nutritional Supplements (Combinations of vitamins (B2, B3, B12, C, E and folic acid).	Increase mitochondrial respiration and simultaneously scavenges free radicals to reduce reactive oxygen species (ROS) and toxic acyl coenzyme A molecules [53].	Effective in combination with Idobenem [54] but effectiveness used as monotherapy in patients with Leber hereditary optic neuropathy (LHON) remains limited and variable [53,55].
2. Brimonidine	Topical a2-agonist. Neuroprotective action in a variety of animal models [56].	Failed in a clinical trial for prophylaxis for second eye involvement in LHON carriers [57].
3. Cyclosporin A	Has an antiapoptotic effect by holding the mitochondrial permeability transition pore closed [58].	It was thought that these drugs may be beneficial in the early stages of LHON by modifying the natural disease progression but study showed that they did not prevent second-eye involvement [59].
4. Phytoestrogens	Targeting estrogen receptor b improve cell viability by reducing apoptosis, inducing mitochondrial biogenesis and strongly reducing the levels of ROS in LHON cells [60].	Treatment should prevent the loss of vision in unaffected individuals carrying the mutation. Studies were made only in vitro, and this method remains experimental pending further evidence [60].
5. Coenzyme Q10 (available as a nutritional supplement)	Ubiquinone, known as Coenzyme Q10, is a lipophilic electron carrier and endogenous antioxidant found in all cellular membranes.	Only a few case reports of improvement of visual acuity have been published [61,62].
6. Gene therapy	Intravitreal delivery of gene therapy when defective gene is replaced with a normal wild-type gene.	Long-term study demonstrated further the potential of gene therapy as a promising safe and effective treatment option. Further studies needed [52].
7. Stem Cells	Utilizes autologous bone-marrow- derived stem cells to treat optic nerve and retinal diseases.	The Stem Cell Ophthalmology Treatment Study (SCOTS) showed that patients with LHON had visual acuity gains and visual field improvement without serious complications. However, further research is needed to examine the role of stem cells in the treatment of LHON [51].

Idebenone is the first and the only approved medication for LHON treatment. Idebenone (Raxone) was approved by European Medicines Agency (EMA) in September of 2015. The recommended dose is 900 mg/day (300 mg, 3 times a day). It is an antioxidant which is capable of transferring electrons directly to complex III of the mitochondrial electron transport chain, thereby circumventing complex I (which is damaged due to LHON-causing mtDNA mutations) and restoring cellular energy (ATP—adenosine triphosphate) generation in the cells [43]. In accordance with this biochemical mechanism, Idebenone may re-activate viable but inactive retinal ganglion cells in LHON patients. This was confirmed by in vivo testing [44]. According to international LHON consensus [48], Idebenone treatment not only prevents regression of the visual function but also improves it.

## 7. Case Report

At the age of 21 male patient of Lithuanian origin presented with sudden painless central vision loss in both eyes. At that time visual acuity (Snellen chart, Landolt C optotype) was 0.3 in both eyes. At the age of 25 based on increased levels of chloride in the sweat, recurrent respiratory infections caused by pathogens specific for cystic fibrosis (*P. Aeruginosa, S. Aureus*), insufficient respiratory function and positive tests for fecal elastase, patient was diagnosed with cystic fibrosis (CF). Later, a genetic molecular research using a sequencing denaturing gradient gel electrophoresis (DGGE) fluorescent-polymerase chain reaction method searching for cystic fibrosis transmembrane conductance regulator (CFTR) gene was conducted, which was negative. At the age of 27 chloride levels in the sweat were almost normal, a gene responsible for the CF was not yet found and so a multidiscipline team of specialists made a conclusion that the diagnosis of CF can not be determined. In addition, patient was diagnosed with asthma. At the age 4 patient had episodes manifesting with ataxia, headaches, and changes in the electroencephalography (EEG), hence diagnosis of epilepsy is not established yet. Patient is a non-smoker and denies any past and present alcohol or recreational drug use.

There was no relevant family history.

Intraocular pressure (IOP) was normal (right eye—13.3 mmHg; left eye—15 mmHg).

On slit lamp examination, the findings of the anterior segment were within normal limits. Ophthalmoscopy revealed bilateral optic nerve atrophy (Figure 2a,b)

Visual field test revealed bilateral central scotomas (Figure 3a,b). Fundus examination showed bilateral optic nerve atrophy (Figure 2a,b). Visual electrophysiology, visual evoked potentials, were performed which showed diminished latency in the right eye, with an prolonged latency in the left eye.

Optical coherent tomography imaging confirmed diminished retinal nerve fibre layer thickness of all four quadrants bilaterally (Figure 4a,b).

Neurological examination and neuroimaging studies showed slender optic nerves and atrophy of chiasm, without any signs of neuroinflammatory process or compressive lesions (Figure 5).

The usual blood test and inflammatory tests were within normal limit.

During follow up visits no new signs or symptoms were noticed.

During the latest patient visit (2018) we were able to perform genetic studies, whole mitochondrial DNA sequencing was initiated (Figure 6) which revealed variants in both *MT-ND4* and *MT-ND1* genes: *m.11253T>C* (likely pathogenic) and *m.3391G>A* (variant of unclear significance), respectively. Method used: polymerase chain reaction and Next Generation Sequencing (NGS) with Illumina MiSeq, computer analysis: BWA2.1 Illumina BaseSpace, Alamut, mtDNA Variant analyser 1000. Tested genes: *MT-ND1, MT-ND2, MT-ND3, MT-ND4, MT-ND4L, MT-NDS, MT-ND6, MT-CYB, MT-CO1, MT-CO2, MT-CO3, MT-ATP6, MT-ATP8, MT-RNRI, MT-RNR2, MT-TA, MT-TT, MT-TP, MT-TE, MT-TL2, MT-TS2, MT-TH, MT-TR, MT-TG, MT-TK, MT-TD, MT-TS1, MT-TC, MT-TY, MT-TN, MT-TW, MT-TM, MT-TI, MT-TQ, MT-TL1, MT-TF, MT-TV*.

A homoplasmic pathogenic variant NC_012920: *m.11253T>C* p.(Ile165Thr) in *MT-ND4* gene was identified.

Based on the clinical presentation, ophthalmological investigations findings and whole mtDNA sequencing results, the diagnosis of LHON caused by pathogenic variant *m.11253T>C* in *MT-ND4* gene was established.

Visual acuity remained stable until the patient was 25 years old (2018) when a sudden visual recovery was documented (a gain of 3 and 4 lines for the right and left eye, respectively) with no change in structural appearance.

At the age of 26 (5 years since disease onset), 2019 May treatment with Idebenone was initiated with a daily dose of 300 mg 3 times a day. Five months later patient’s visual acuity kept improving with 0.8 (right eye) and 0.9 (left eye).

## 8. Discussion

We report a case of LHON which was caused by a rare point mutation in *MT-ND4* gene. The patient was identified to carry pathogenic *m.11253T>C (MT-ND4)* variant, which previously has been reported in patient with LHON with spontaneous visual recovery [63].

Spontaneous partial visual recovery is rare and it is mostly observed in patients harbouring *m.14484T>C* mutation [19,20,64]. It is known that the progression of LHON for patients with this mutation is milder and the chance of visual improvement is from 37 to 71%. In comparison, patients with *m.11778G>A* and *m.3460G>A* mutations have only a 4% chance of spontaneous recovery [2].

There are some reports of LHON cases which are caused by rare mutations and later spontaneous clinical recovery was noticed. Emperador et al. published a case where LHON was likely to be caused by *m.13094T>C* and *m.15527C>T* mutations in a 10-year-old boy who later experienced spontaneous visual recovery [65].

The *m.11253T>C* mutation which is associated with LHON in our patient, has previously been reported in patient with proven Parkinson’s disease [63], hence our patient’s family history is negative for this neurodegenerative disorder.

To date, the mechanism of spontaneous visual recovery in LHON remains unknown. Recently it was noted that remission is quite common if LHON manifests in childhood [65] but such cases are rare. Ramos et al. in their study showed that relatively bigger optic nerve head could be a prognostic factor for a spontaneous visual recovery in LHON patients [18]. Moon and others carried out a retrospective study where they tried to identify some factors of possible spontaneous visual recovery which was described as a gain of 3 or more lines. 21 of 80 eyes had visual recovery and such patients were younger at LHON onset (21 vs. 29 years old) and had better vision at the nadir. In the study, fundus photos were evaluated, and it was noticed that the presence of peripapillary telangiectasia and optic disc hyperemia may serve as predictive factors for poor visual prognosis in patients with LHON [66]. In comparison, our patient was 25 years old when the visual recovery happened, and the visual acuity was 0.3 at the nadir. Also, there were not any peripapillary telangiectasias or optic disc hyperemia during ophthalmoscopy and optic nerve head size was medium in both eyes.

Our case also showed that treatment with Idebenone could be effective even after spontaneous remission already occurred and atrophy of optic nerve disc developed. Initially Idebenone was only recommended for patients in early LHON stages but Pemp et al. conducted a research which revealed that this treatment is effective even in late stages when optic nerve atrophy had developed. Treatment was started in seven patients 5 to 51 years after LHON onset and there was a significant increase in visual acuity by an average of −0.20 ± 0.10 logMAR or 10 ± 5 ETDRS letters [67]. This proves that Idebenone should be prescribed even if the LHON is found in the late stages.

## Figures and Tables

**Figure 1 medicina-57-00202-f001:**
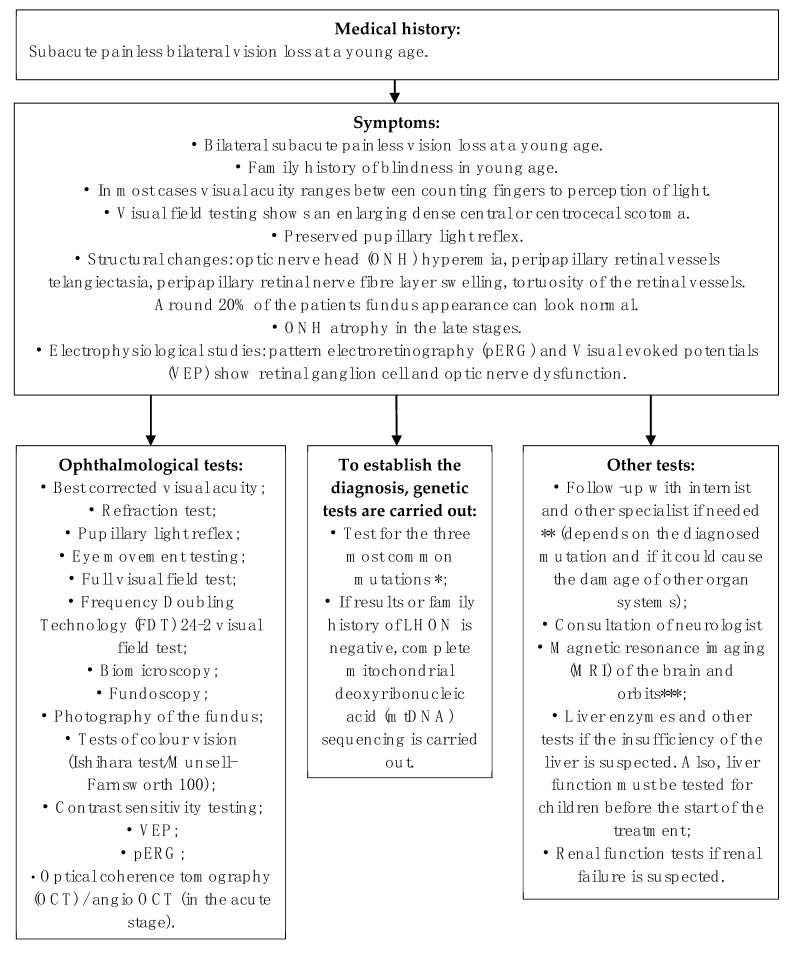
The diagnostic approach of the patient with suspected LHON. * *m.11778G>A*, *m.14484T>C*, *m.3460G>A*. ** Cardiac conduction defects and LHON: a Finnish study showed an increased incidence of cardiac arrhythmias secondary to accessory pathways in association with LHON [33], but this finding was not noticed when other populations were examined [34]. *** Neuroradiological tests: MRI results are often normal but may reveal white matter lesions and/or a high signal within the optic nerves [35]. Also, MRI could show thinned optic nerves and optic tract.

**Figure 2 medicina-57-00202-f002:**
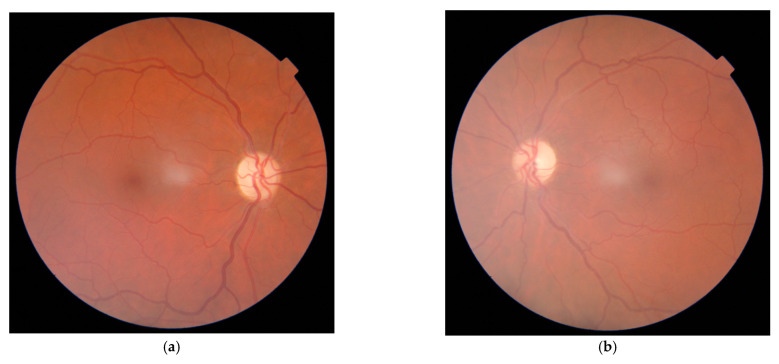
(**a**,**b**) Bilateral optic nerve atrophy, no changes in the central and peripheral retina.

**Figure 3 medicina-57-00202-f003:**
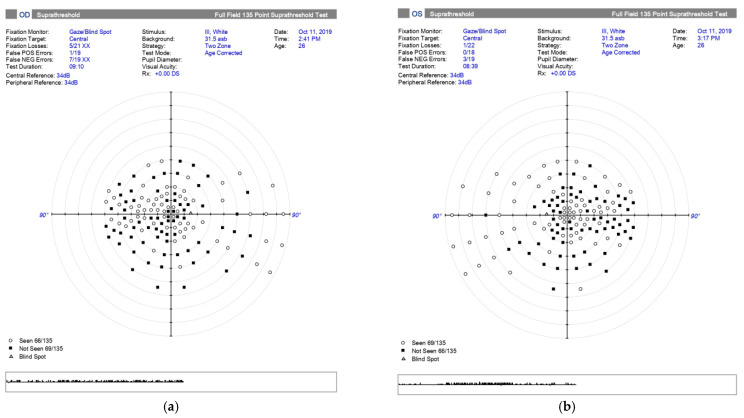
Visual field testing at the time of sudden painless central vision loss showed central scotomas in both eyes (**a**) Right eye—not seen points: 66/135; (**b**) Left eye—not seen points: 69/135.

**Figure 4 medicina-57-00202-f004:**
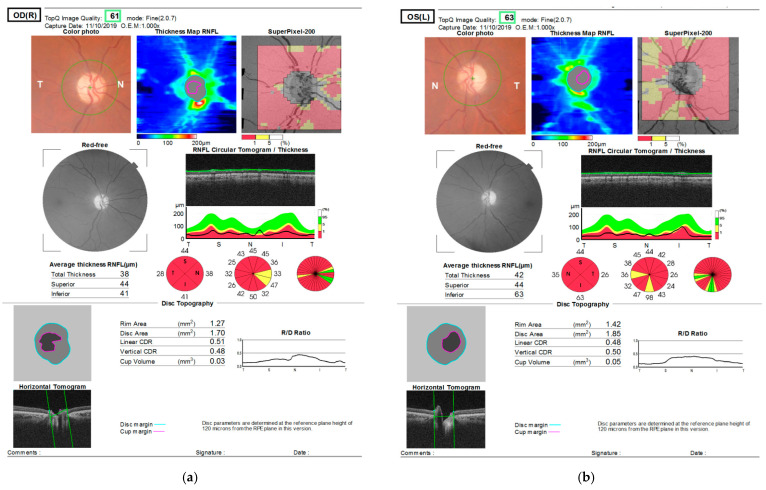
Optical coherent tomography shows thinning of retinal nerve fibre layer (RNFL) in both eyes. (**a**) right eye: nasal quadrant—38 µm, inferior 41 µm, temporal 28 µm and superior 44 µm; (**b**) left eye: nasal quadrant—26 µm, inferior 63 µm, temporal 35 µm and superior 44 µm.

**Figure 5 medicina-57-00202-f005:**
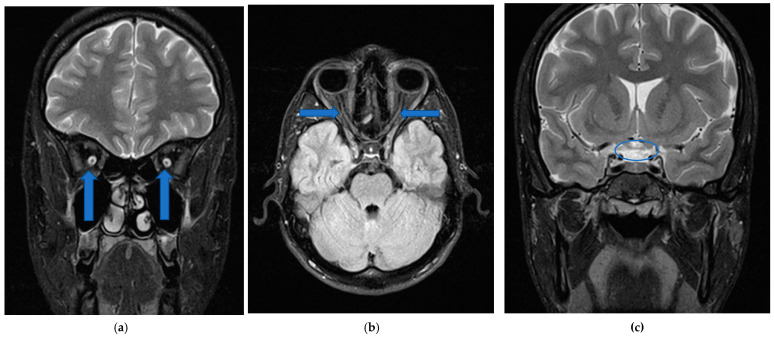
1,5T magnetic resonance imaging (MRI) neuroimaging study showed no signs of neuroinflammatory process or compressive lesions, but visualize (**a**) (T2_TSE sequences) and (**b**) (FLAIR_longTR sequences) slender optic nerves (**c**) (T2_TSE sequences) atrophy of chiasm.

**Figure 6 medicina-57-00202-f006:**
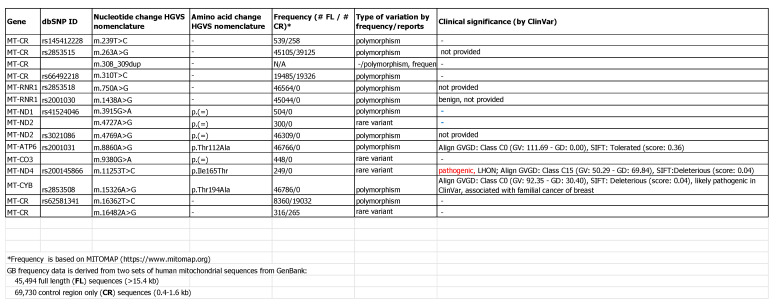
Mitochondrial DNA sequencing results.

## Data Availability

All data could be seen in this manuscript.

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
