# Peer review of "A Typical Case Presentation with Spontaneous Visual Recovery in Patient Diagnosed with Leber Hereditary Optic Neuropathy Due to Rare Point Mutation in MT-ND4 Gene (m.11253T>C) and Literature Review"

_medicina, 2021, doi:10.3390/medicina57030202_

Round 1

Reviewer 1 Report

The authors presented basic information about the genesis, course and clinical picture of Leber's disease, as well as a case study of a patient who experienced a spontaneous (or rather unexplained) improvement in vision.
In my opinion, the case presentation is insufficient and largely consists of measurement data. However, there is no description of the entire background, e.g. related to possible coexisting diseases, family burden, treatment of the disease itself, etc., and most of all, no attempt was made to explain the spontaneous resolution of symptoms. Taking into account the fact that there is a lot of data on the disease itself in the textbooks, the scientific-clinical publication should be much more focused on a very advanced and multifaceted characterization of the case itself. There is absolutely no discussion of the material presented. 

Besides, the following issues should be addressed: 

- Lack of literature references at the beginning of disease description in point 3.

- Figure 4 should be presented with bigger details. 

- There are several incomprehensible sentences, such as: "We Neurological examination and neuroimaging studies showed slender optic nerves and atrophy of chiasm, without any signs of neuroinflammatory process or compressive lesions." (pp. 7)

- Neuroimaging pictures should be given in better solution and with more detailed indicators of localized pathology. No technical details were given regarding neuroimaging methods used. It should be noted that many of the recent 7T MRI studies in Leber's disease have shown many structural abnormalities in the disease, possibly imperceptible in a 1.5 T MRI study. Please refer to this.

- What was the exact time lapse between visits?

Author Response

The rows we numbered according to the document version “Track changes – All markup”

Answers to the comments made by 1st reviewer:

  • We added information about the case and the patient which could be found in the lines 215-232 and 286-293.
  • We wrote a discussion which is presented in line 294-332.
  • We added references to the statements at the beginning of disease description in point 3 (lines 79-133)
  • We presented a bigger figure 4 so its details are now more visible.
  • We corrected incomprehensible sentences (line 215 and 259)
  • Images of MRI are now added, and localized pathology is shown in them (figure 5). This study of MRI was a 1,5T because a 7T MRI was not possible at the time. Patient also had a 3T MRI study in other clinic but the results were the same as in a study which we provided in the article.
  • We added exact time of each visit in case report section (line 215, 218, 224, 268, 290).

Reviewer 2 Report

This is a short overwiev of data regarding molecular and clinical aspects of LHON with presentation of an unusual case caused by m.11253T>C mutation.

It would be worth to add a precise description of a method of genetic analysis (Sanger seq or NGS, what equipment) and to show the results of sequencing presenting at least the above mentioned mutation. It would be also important to inform what is the level of heteroplasmy. 

Author Response

The rows we numbered according to the document version “Track changes – All markup”

Answers to the comments made by 2nd reviewer:

  • We added a description of genetic analysis method and its results in lines 271-279. We also added a figure 6 which represents the results of genetic testing.
  • The level of heteroplasmy is mentioned in the line 278.

Reviewer 3 Report

In this manuscript, the authors reported a case of patient with Leber hereditary optic neuropathy (LHON) harbouring a rare point mutation in MT-ND4 gene, also giving an overview of the literature. Although interesting, I think that the manuscript requires some revisions to get it acceptable for publication:

  • Rewrite the paragraph regarding the aetiology. I think that it’s too simplistic and not “scientifically” described.
  • What do you mean for “exotic” mutations? Can you report some examples?
  • Are there any comorbidities with LHON?
  • Regarding the treatments, I should suggest the authors to include in the table the pros and cons of each therapeutic options, if available. Moreover, they should also speculate on possible new therapies/targets.
  • Please insert a Conclusion section criticising and summarising the described findings
  • Please put the name gene in italics 

Author Response

The rows we numbered according to the document version “Track changes – All markup”

Answers to the comments made by 3rd reviewer:

  • We rewrote the aetiology which is now new information could be found in lines 47-77.
  • We deleted the sentence about the “exotic” mutations because we used in the wrong manner. The following changes about rare mutations were made and now could be found in lines 52-55.
  • We added information about the patient comorbidities in case report section, line 218-230.
  • We added some information about the non-approved treatment options in the Table 1 and 197-199 lines. We believe that this figure is informative as it is and some pros and cons are mentioned but in general it is way too early to determine them because of the lack of studies regarding these methods.
  • We rewrote gene names in italic.

Round 2

Reviewer 3 Report

The authors have done their best to address all my raised issues. This is a much improved revised version of the previously submitted manuscript.